# D-MannosE to prevent Recurrent urinary tract InfecTions (MERIT): protocol for a randomised controlled trial

Marloes Franssen [ID],[1] Johanna Cook,[2] Jared Robinson,[2] Nicola Williams,[2] Margaret Glogowska,[2] Yaling Yang,[2] Julie Allen,[2] Christopher C Butler [ID],[2] Nick Thomas,[3] Alastair Hay,[4] Michael Moore [ID],[5] Gail Hayward [ID][2]

► Prepublication history and supplemental material for this paper is available online. To view these files, please visit the journal online (http://dx.doi.org/10.1136/bmjopen-2020-037128).

[1]Nuffield Department of Orthopaedics, Rheumathology and Musculoskeletal Sciences, University of Oxford, Oxford, UK
[2]Nuffield Department of Primary Care Health Sciences, Oxford University, Oxford, UK
[3]Windrush Medical Practice, Witney, UK
[4]Centre for Academic Primary Care, University of Bristol, Bristol, UK
[5]Primary Care Medical Group, University of Southampton, Southampton, UK

**Correspondence to**
Dr Gail Hayward;
gail.hayward@phc.ox.ac.uk

## ABSTRACT

**Introduction** Recurrent urinary tract infections (RUTIs) have a significant negative impact on quality of life and healthcare costs. To date, daily prophylactic antibiotics are the only treatment which have been shown to help prevent RUTIs. D-mannose is a type of sugar which is believed to inhibit bacterial adherence to uroepithelial cells, and is already being used by some women in an attempt to prevent RUTIs. There is currently insufficient rigorous evidence on which to base decisions about its use. The D-mannose to prevent recurrent urinary tract infections (MERIT) study will evaluate whether D-mannose is clinically and cost-effective in reducing frequency of infection and symptom burden for women presenting to UK primary care with RUTI.

**Methods and analysis** MERIT will be a two-arm, individually randomised, double blind placebo controlled, pragmatic trial. Participants will be randomised to take D-mannose powder or placebo powder daily for 6 months. The primary outcome will be the number of medical attendances attributable to symptoms of RUTI. With 508 participants we will have 90% power to detect a 50% reduction in the chance of a further clinically suspected UTI, assuming 20% lost to follow-up. Secondary outcomes will include: number of days of moderately bad symptoms of UTI; time to next consultation; number of clinically suspected UTIs; number of microbiologically proven UTIs; number of antibiotic courses for UTI; quality of life and healthcare utilisation related to UTI. A within trial economic evaluation will be conducted to examine cost-effectiveness of D-mannose in comparison with placebo. A nested qualitative study will explore participants' experiences and perceptions of recruitment to, and participation in a study requiring a daily treatment.

**Ethics and dissemination** Ethical approval has been obtained from South West-Central Bristol Research Ethics Committee. Publication of the MERIT study is anticipated to occur in 2021.

**Trial registration number** ISRCTN 13283516.

### Strengths and limitations of this study

► Based on current literature, this will be the first large publicly funded randomised controlled trial of D-mannose for prophylaxis of recurrent urinary tract infections.
► This study is the first to use a placebo control in evaluating the benefit of D-mannose.
► Obtaining the primary outcome by medical notes review will ensure data completeness.
► The trial may not be powered to detect a secondary outcome of symptom burden which is also of value to patient decision making.
► Although participants report weekly on their study product usage there are no objective measures available to confirm accuracy of reporting.

UTIs (RUTIs) have a considerable negative impact on quality of life, which extends beyond the unpleasant symptoms to distressing and disrupted sexual relationships, persistent unmanageable pain and systemic illness.[4] UTI accounts for an important proportion of healthcare costs as a result of outpatient visits, diagnostic tests and prescriptions.[5] In 2007, UTI recurrence accounted for 10.5 million outpatient consultations and 2–3 million emergency department visits in the USA alone. In addition, UTIs are the most common cause of infection in hospitalised patients, accounting for 17.2% of all nosocomial infections in England. Furthermore, UTIs result in considerable patient morbidity and time off work; hence, the management of this condition incurs large financial costs, estimated at US$3.5 billion in the USA per year.[6]

A systematic review of randomised controlled trials identified antibiotic prophylaxis as the only treatment, which has been demonstrated to help prevent RUTIs. Antibiotics taken daily for 6–12 months were more effective than placebo at preventing

## BACKGROUND

Urinary tract infection (UTI) is the most common bacterial infection that women consult for in UK primary care.[1 2] Approximately 40%–50% of women experience one UTI episode during their lives.[3] Recurrent

recurrent infection,[7] and national guidelines advocate their use for this indication.[8] However, antibiotics also resulted in more severe and unpleasant side effects (eg, vomiting, urticaria, candidiasis). Furthermore, once antibiotic prophylaxis is discontinued, even after extended periods, approximately 50%–60% of women will experience a further UTI within 3 months.[9 10] Thus, antibiotic prophylaxis does not exert benefit once stopped, and is directly linked to antibiotic resistance in uropathogens.[11] Antibiotic resistance has been associated with an increased duration of severe symptoms of UTIs, irrespective of the use of an appropriate antibiotic.[2 11]

D-mannose is a type of sugar (a monosaccharide isomer of glucose), which is thought to inhibit bacterial adherence to uroepithelial cells by binding to a site on the tip of the fimbria[12] and has shown benefit in animal models in preventing UTIs.[13]

Currently D-mannose is available commercially to the public as a food supplement, and is favoured by many women who have RUTIs, but until recently, there has been little empirical evidence to support its use. An open-label randomised three arm trial including 308 women with RUTI seen in outpatient settings[14] found that daily use of D-mannose for 6 months resulted in an absolute reduction in incidence of further UTI of 45% from a proportion of 60% in the usual care arm, with no adverse events. The proportion of women experiencing an RUTI over 6 months was reduced by 11% compared with daily antibiotic use. This finding is supported by recent smaller studies.[15–18]

Although there are indicators of efficacy from small underpowered trials, the only adequately powered study to date[15] was not placebo controlled and found an unexpectedly high RUTI incidence in the control arm. Furthermore, a microbiologically confirmed UTI was a requirement for entry to the study, and participants were withdrawn once they developed a UTI on treatment. Therefore, true incidence of UTI could not be established, a measure for women who experience frequent RUTIs, who are also the most likely candidates for prophylaxis. Finally, all women on hormonal contraception were excluded, which may reduce applicability to the women at high risk of RUTI.

D-mannose is found naturally in small quantities in numerous food sources, such as coffee, baker's yeast, egg white, fruits such as apples, cranberries and mangos, and also in legumes such as soybeans, kidney beans and peanuts.[19] It is absorbed in the upper gastrointestinal tract and excreted in the urine.[14]

D-mannose may offer an alternative to antibiotic prophylaxis in women who experience RUTI and in turn to contribute to better antimicrobial stewardship in primary care. However, the current evidence base is inadequate to help women with RUTI to make informed decisions about the use of D-mannose prophylaxis. The high costs (at least £25 a month) associated with its purchase add weight to the need to establish whether general practitioners (GPs) should advise their patients to use D-mannose for this indication.

The D-mannose to prevent recurrent urinary tract infections (MERIT) double blind placebo-controlled randomised controlled trial aims to evaluate the effectiveness of D-mannose in women suffering with RUTI presenting to UK primary care and its cost effectiveness.

## METHODS AND DESIGN
### Study aims, research questions and outcomes
The primary aim of MERIT is to assess the effectiveness of daily use of D-mannose compared with placebo in preventing symptomatic UTI in women.

The primary outcome of the trial will be the proportion of women experiencing at least one further episode of clinically suspected UTI for which they contact ambulatory care (out of hours primary care, in hours primary care, ambulance or the emergency department) within 6 months of study entry.

Secondary outcomes will include (within 6 months of study entry):
► Number of days of moderately bad (or worse) symptoms of UTI.
► Time to next consultation with a clinically suspected UTI.
► Number of clinically suspected UTIs.
► Number of microbiologically proven UTIs.
► Number of antibiotic courses for UTI.
► Report of consumption of antibiotics using diary during periods of infection.
► Proportion of women with a resistant uropathogen culture during an episode of acute infection.
► Hospital admissions related to UTI.
► Quality of life and healthcare utilisation related to UTI.
► Healthcare utilisation recorded in the participant diary and during a notes review.
► Acceptability and process evaluation conducted via telephone interviews (after 6 months).

### Study design and setting
A two-arm, individually randomised, double blind placebo controlled, pragmatic trial. At least 50 GP practices in England and Wales will be invited to take part in the trial. Recruitment will run from March 2019 to January 2020.

### Eligibility
This trial will recruit female participants over 18 years with a primary care clinical record of having presented to ambulatory care with RUTIs three or more times in the last year or two or more times in the last 6 months. Exclusion criteria are: participants who are pregnant, lactating or planning pregnancy during the course of the study; formal diagnosis of interstitial cystitis or overactive bladder syndrome; prophylactic antibiotics started in the last 3 months and unwilling to discontinue, or intention to start in the next 6 months; currently using D-mannose

## Box 1   Data collection throughout the trial

Baseline:
1. Demographic questions: including age.
2. Medical history (by patients).
3. Use of contraceptives and hormonal treatment.
4. Urinary tract infection (UTI) episodes in the last 12 months.
5. EuroQol 5-dimensional questionnaire (EQ-5D-5L).
Weekly contact/monthly contact:
1. UTI episodes in last week/month, respectively.
Daily UTI diary
1. UTI symptoms.
2. UTI treatment.
3. EQ-5D-5L.
Six month questionnaire
1. UTI episodes in the last month.
2. EQ-5D-5L.
Notes review and urine culture result
1. Recorded UTIs during the study period.
2. Healthcare contact for UTIs recorded.
3. Antibiotics given for UTIs recorded.
4. Culture results for UTIs recorded.
5. Unscheduled hospital admissions.

and unwilling to discontinue for the duration of the study; nursing home resident; catheterised, including intermittent self-catheterisation; use of uromune; participation in a research study involving an investigational product in the past 12 weeks.

### Baseline assessment

Participants will have a baseline assessment (either in person with their GP or research nurse online, or by telephone with a member of the research team). During the baseline assessment the study will be explained, informed consent (see online supplemental file 1) will be obtained and data will be collected (see box 1). Participants will also be asked to send in a urine sample (when they are asymptomatic of a UTI) at baseline and have the option to also send in a perineal swab sample.

### Randomisation

After the baseline assessment, participants will be randomised by a member of the research team using a validated internet based randomisation system with an emergency randomisation list available. Randomisation will use variable block sizes and will be stratified by GP practice ensuring a balance of the two arms within each practice.

### Intervention and placebo groups

Placebo will consist of two grams of a sugar powder which is similar in texture and taste to D-mannose but fully absorbed by the liver to be taken daily for 6 months.

Intervention will consist of two grams of D-mannose powder to be taken daily for 6 months.

An adequate supply of the study product will be sent directly from the research team to the participant after randomisation, after 2 months and after 4 months.

### Follow-up

All participants will be asked to complete short weekly questionnaires, sent to them via text or email; they also will have the option of completing them telephonically directly with the research team. The weekly questionnaire will collect the information of participant's adherence to study medication, and whether the participant has had any symptomatic UTI episodes. Participants will also be contacted monthly by phone by the research team to complete a monthly questionnaire which is similar to the weekly one if there are two or more weekly questionnaires not being completed. They will also be asked to complete a daily UTI symptom diary if they experience a UTI. The information will be collected via weekly and monthly contact if they fail to complete the symptom diary although they experience one. During a UTI they will be asked to send the lab a urine sample, alongside any sample they might provide to their GP. They will also be asked for a further urine sample 2 days after symptoms have resolved. See box 1 for details. Primary care electronic medical record reviews will be conducted to collect UTI related healthcare contacts, culture results and prescriptions during the following up period. Participants will receive a £10 voucher after every 2 months of participation (£30 pounds in total). See figure 1 for the participant flow through the trial.

### Sample size considerations

A recent study to evaluate prophylactic treatment for RUTI in a similar population[20] found that 26.6% of women in the control arm experienced an RUTI within 6 months. Our patient and public involvement advisors suggested that in order to commit to daily use of a prophylactic regime, they would require evidence of at least a 50% reduction in the chance of a further UTI during the period of prophylaxis. To detect this reduction with a two-sided Fisher's exact test with 90% power and an alpha of 0.05 we would require 203 participants in each arm. This equates to 508 participants if a 20% loss to follow-up is assumed. This sample size is also adequate to power the key secondary outcome (the number of RUTI's experienced over 6 months), and detect a relative incidence rate of 0.5 between the treatment and placebo groups, assuming a base rate of 0.36 as estimated by Maki et al.[20] If the estimated percentage of participants who have either withdrawn or failed to respond to any study team communication for an extended period seems likely to rise above the 20% initially allowed for, we will recruit additional participants, up to a maximum of 598 participants.

### Statistical analysis

The primary outcome, the proportion of women experiencing at least one further episode of UTI symptoms for which they visited their GP within 6 months of study entry, and other binary outcomes, will be analysed on an intention-to-treat basis by means of a generalised linear mixed effects model with binomial distribution and log link function, including a random effect for practice and

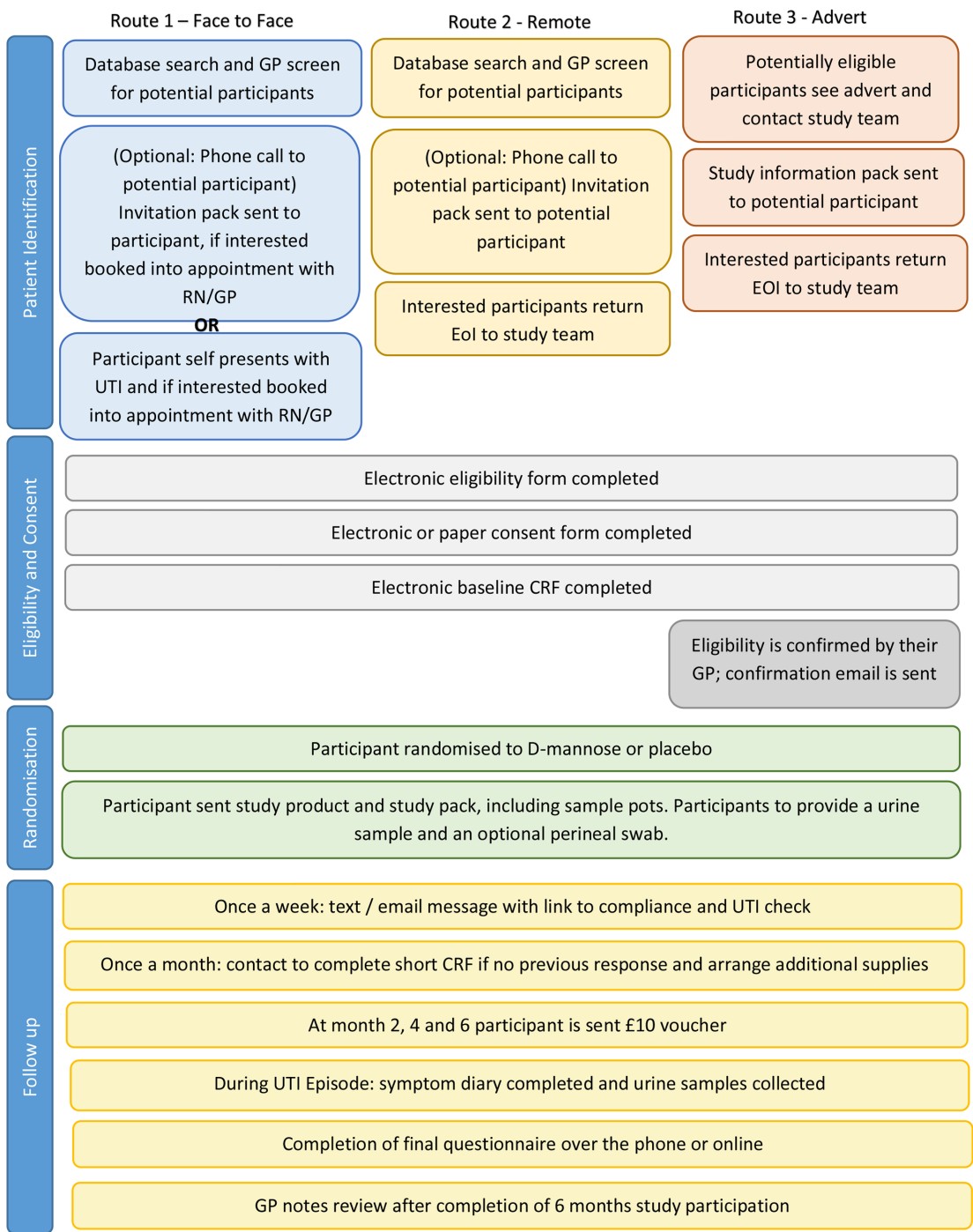

**Figure 1** Flow through the trial. CRF, case report form; EOI, expression of interest; GP, general practitioner; RN, research nurse; UTI, urinary tract infection.

fixed effect for randomisation group. Therefore, treatment groups will be compared on the basis of an adjusted risk ratio. The number of days of moderately bad symptoms of UTI, the number of UTI's experienced in 6 months, and number of antibiotic courses for UTI in 6 months, will be analysed by means of a generalised linear mixed effects model using the Poisson distribution and log link function, including a random effect for practice and a fixed effect for randomised group. Defined daily ooses (DDDs) will be analysed by means of a linear mixed-effects model including a random effect for practice and

fixed effects for randomised group and baseline DDD, treating this outcome as continuous. We will analyse the overall DDD as well as the individual antibiotic DDDs.

The amount of missing primary outcome data is expected to be very low as it is collected via notes review. The model chosen to analyse the primary outcome implicitly accounts for data missing at random, however, the data missing mechanism will be explored. Summary statistics will be presented for baseline covariates of those participants who completed and those who were lost to follow-up for the primary outcome. Baseline covariates

associated with missingness will be identified by analysing each baseline covariate in a logistic regression model to determine which (if any) are associated with missingness of the primary outcome. The associated p value will be reported alongside the summary statistics. Any baseline factors found to be associated with missingness of the primary outcome will be included in a sensitivity analysis.

### Data management
Data management will be performed in accordance with Primary Care Clinical Trials Unit Data Management standard operating procedures. Study-specific procedures will be outlined in a Data Management Plan to ensure that high quality data are produced for statistical analysis.

### Potential risks
It is anticipated that the potential risks of this study are low and similar to those attributable to usual care.

### Health economic evaluation
A cost effectiveness analysis from a health system perspective with a time horizon of 6 months will be conducted alongside this study. The primary outcome measure for the cost utility analysis will be the quality adjusted life years (QALYs). Data collection to facilitate analysis includes resource use and health outcomes. Data from the participant diary and electronic medical record review will be the main source of resource use. Unit costs associated with resource use items will be obtained from national standards. Health outcomes will be measured using the 5-level version of the EuroQol 5-dimensional questionnaire (EQ-5D-5L).

Data analysis will be conducted on an intention-to-treat basis using an incremental approach. Resource use and unit cost will be combined to calculate healthcare costs for each participant and mean cost for each study arm. EQ-5D-5L utility values will be calculated using the UK-based algorithm. Using the under the curve methods to combine utility values and associated time durations will produce QALYs for each participant and mean QALYs for each study arm during the 6-month study period. Mean differences in costs and QALYs between the study arms will be estimated as incremental cost per QALY gained. Given the fact that antibiotics are currently the mainstay treatment for both acute and RUTIs, the issue of how the cost of antibiotic resistance should be incorporated into economic evaluation will be explored in the analysis.

### Nested qualitative study
We will recruit a maximum variation sample of 35 participants across both study arms for the nested qualitative study, continuing recruitment until data saturation is reached. A balanced list of participants will be drawn up for the qualitative researchers. The topic guide will include participants' experiences and perceptions of recruitment to, and participation in a study that requires taking a daily study product (whether D-mannose or placebo), exploring the level of perceived benefit patients

anticipate would be required for them to continue this type of regimem, and facilitators and barriers to adhering to prophylactic treatment. For participants' convenience, interviews will be conducted by telephone. Thematic analysis of the interviews will take into account issues identified from the literature and clinical research context, as well as inductively allowing new themes and ideas to emerge from the data. Analysis will be guided by the constant comparative method,[21] which will include reading and familiarisation with the transcripts, noting and recording initial themes and then conducting systematic and detailed open coding using NVivo V.12,[22] a qualitative data analysis software. Analysis will proceed in an iterative manner—thus, the coding of a first set of interviews will generate an initial coding framework, which will be further developed and refined as further interviews are conducted and analysis proceeds. The researcher will draw on the clinical expertise of the rest of the research team in developing the coding framework and critically discussing ideas for categories emerging from the data, to ensure trustworthiness. A reflexive journal will assist in interpreting data and forming conclusions.

### Patient and public involvement
Members of the public were involved in the design of the trial, reviewed patient facing documents and they will be active members of the trial steering committee.

## DISCUSSION
The MERIT study will be the first large, publicly funded, double blind randomised trial of the clinical and cost-effectiveness of daily D-Mannose for preventing RUTI in primary care. This overview of the protocol describes the plans for a pragmatic study recruiting women who suffer from RUTI recruited in UK primary care. This study will fill a major gap in the evidence base about whether women with RUTIs should initiate or continue to use this food supplement to prevent RUTI. If D-mannose is proven to be effective for the treatment of RUTIs this could benefit affected women and also contribute to antimicrobial stewardship. On the other hand, if found to be ineffective, costs spent on an ineffective intervention will be saved and attention can be refocussed on other, perhaps more effective prophylactic approaches, as well as redirected research efforts.

### Ethics and dissemination
Ethical approval has been obtained from South West-Central Bristol Research Ethics Committee (reference: 18/SW/0245). Any subsequent protocol amendments will be agreed with both sponsor and ethics committee prior to implementation. Publication of the MERIT study is anticipated to occur in 2021.

**Acknowledgements** The authors acknowledge the support of the Primary Care Clinical Trials Unit. Patient representatives are Sylvia Bailey and Valerie

Tate. Additional members of the TSC are Rebecca Cannings-John (chair), Laura Shallcross and Akke Vellinga.

**Contributors** GH had the original idea with CCB and they designed the study and obtained the funding. MF and JC wrote the first draft, NW provided the statistical section. YY provided the health economic section, MG provided the qualitative section. All authors (JR, JA, NT, AH, MM) subsequently critically edited the manuscript. GH will be guarantor for the manuscript.

**Funding** The trial is funded by a School for Primary Care research grant. Service support costs are administered through the NIHR Clinical Research Network: Thames Valley and South MidlandsThis article presents independent research commissioned by the National Institute for Health Research (NIHR) under a School for Primary Care Research Grant (reference number:385). The views expressed in this publication are those of the authors and not necessarily those of the NHS, the NIHR or the Department of Health.

**Disclaimer** The results of the MERIT will be published in peer-reviewed journals, in addition to being presented at conferences. It is anticipated that the results will be published in 2021, after the 6 months follow-up of all recruited participants. The sponsor and funder had no role in the study design; collection, management, analysis, and interpretation of data, writing of the report; or the decision to submit the report for publication, which was made jointly by the authors who have all approved the final manuscript.

**Competing interests** None declared.

**Patient consent for publication** Not required.

**Provenance and peer review** Not commissioned; externally peer reviewed.

**ORCID iDs**
Marloes Franssen http://orcid.org/0000-0002-1134-7929
Christopher C Butler http://orcid.org/0000-0002-0102-3453
Michael Moore http://orcid.org/0000-0002-5127-4509
Gail Hayward http://orcid.org/0000-0003-0852-627X

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
