## [Reviewer comments · BMJ Open]

ARTICLE DETAILS

TITLE (PROVISIONAL)	D-mannose to prevent recurrent urinary tract infections (MERIT): protocol for a randomised controlled trial.
AUTHORS	Franssen, Marloes; Cook, Johanna; Robinson, Jared; Williams, Nicola; Glogowska, Margaret; Yang, Yaling; Allen, Julie; Butler, Christopher C.; Thomas, Nick; Hay, Alastair; Moore, Michael; Hayward, Gail

VERSION 1 – REVIEW

REVIEWER	Prof. Abdulbari Bener Professor of Public Health Dept. of Biostatistics & Medical Informatics Cerrahpaşa Faculty of Medicine Istanbul University Cerrahpaşa and Istanbul Medipol University, International School of Medicine 34098 Cerrahpasa-Istanbul, TURKEY
REVIEW RETURNED	22-Feb-2020

GENERAL COMMENTS	Overall, this study is well written and well designed and it addresses an important public health issue. Although, the study does not contribute novel knowledge or add sufficiently to the current literature but can be adopted in other countries and it would help local policy makers,
---

REVIEWER	Aaron Wendelboe University of Oklahoma Health Sciences Center
REVIEW RETURNED	22-Feb-2020

GENERAL COMMENTS	The study protocol for "D-mannose to prevent recurrent urinary tract infections (MERIT)" is well-written and the study should have the intended impact. There are a few areas, as detailed below, in which the authors may improve clarity. Background: Do you have any information on how much money is spent on D-mannose and similar supplements? There was a citation of costing 25 pounds per bag, but what about an estimate of the industry? Page 6, line 47. Should it be "...women experiencing a recurrent UTI..."? Please scan throughout the text. I believe there are a number of areas where you refer to a patient's UTI, I assume that recurrent UTI is the intended meaning, but it's not clear. (Page 7, line 26 is another instance) Page 6, line 53. This is a long complicated sentence. Please break into two sentences. Page 7, line 31. What is A&E?
---

	Page 7, line 56. The section heading is Study Design and Setting, but there is only a single sentence describing the study design. Please provide additional information about the setting. Information about participating GP's is particularly needed. Page 8, section 2.4. Please describe the incentive, if any for patients' participation. (There is mention of an incentive in Figure 1, but it should also be included in the text of the protocol.) Page 8, line 39. There is a typo after "liver" Page 8, line 57. Additional details need to be included about how many missed weekly and monthly reports results in action being taken. Page 9, line 3, edit to "They will also be asked..." Sections 2.12 and 2.13: I do not feel qualified to review these sections. Page 10, line 34, should it be "type of regimen"? From a brief literature search, it appears there are quite a few recent relevant articles the authors may want to consider incorporating.
--	--

REVIEWER	Wagenlehner Florian Justus Liebig University Giessen, Germany
REVIEW RETURNED	31-May-2020

GENERAL COMMENTS	The study protocol is well designed and able to address the research question. In the sample size calculation paragraph the authors rely to a publication of another group investigating recurrent UTI, where in the control arm only 26% of included patients had at least one UTI in the 6 months follow up. If the definition of recurrent UTI would be applied, one would expect that 100% of the patients in the control arm would have at least 2 UTIs in the 6 months follow up. How would this discrepancy impact the study?
--

REVIEWER	John Stephenson University of Huddersfield, UK
REVIEW RETURNED	11-Aug-2020

GENERAL COMMENTS	My review focuses on the statistical aspects of the protocol only. I would like to request the following points of clarification.  1. I was able to reproduce the sample size calculation (section 2.8) when using the Fisher exact test. I assume that is what the authors used to generate their figures. It would have been helpful if they had stated that this was the method they were using, rather than any of the other possible options including options such as uncorrected chi square which are more commonly used than the Fisher test. 2. Please can the authors explain why they are comparing treatment groups on the basis of an adjusted risk ratio (section 2.9) rather than an adjusted odds ratio. 3. Please can they also explain why they are using a log link function (section 2.9) for the primary outcome and not a logit link or similar as I would have expected to see for a proportion. 4. Section 2.5: "Randomisation will be stratified by GP practice ensuring a balance of the two arms within each practice." Presumably some form of block randomisation will be utilised to "ensure balance" - but no details are given. How large are the blocks? Are block sizes randomised?
--

	5. It would be interesting to know why it is necessary to stratify by practice. The implication is that there is some key practice-based factor which it is essential to distribute equally across groups. Achieving this comes at a cost (more elaborate analytical methods needed; less uncertainty in pre-determination of allocation. Please could the authors explain why they felt that this procedure was necessary. What is the factor that has to be balanced across groups? With a sample size of over 500, what is the risk of this not happening using blocked, but unstratified randomisation? 6. What is the plan for missing data, in particular missing outcome data? I couldn't find that. The researchers state that they will be conducting an ITT analysis but this can be highly problematic with missing data.
--	--

VERSION 1 – AUTHOR RESPONSE

Reviewer(s) Reports:

Prof. Abdulbari Bener

Professor of Public Health

Dept. of Biostatistics & Medical Informatics Cerrahpaşa Faculty of Medicine Istanbul University

Cerrahpaşa and Istanbul Medipol University, International School of Medicine

34098 Cerrahpasa-Istanbul,

TURKEY

Please state any competing interests or state 'None declared':

'None declared':

Overall, this study is well written and well designed and it addresses an important public health issue. Although, the study does not contribute novel knowledge or add sufficiently to the current literature but can be adopted in other countries and it would help local policy makers,

Reviewer: 2

Aaron Wendelboe

University of Oklahoma Health Sciences Center

Please state any competing interests or state 'None declared':

None declared

The study protocol for "D-mannose to prevent recurrent urinary tract infections (MERIT)" is well-written and the study should have the intended impact. There are a few areas, as detailed below, in which the authors may improve clarity.

Background: Do you have any information on how much money is spent on D-mannose and similar supplements? There was a citation of costing 25 pounds per bag, but what about an estimate of the industry? Thank you. This information is unfortunately not available.

Page 6, line 47. Should it be "...women experiencing a recurrent UTI..."? Please scan throughout the text. I believe there are a number of areas where you refer to a patient's UTI, I assume that recurrent UTI is the intended meaning, but it's not clear. (Page 7, line 26 is another instance) Thank you, updated on page 4 line 19; page 6 line 47; page 6 line 52; page 6 line 57; page 9 line 13; page 9 line 21

Page 6, line 53. This is a long complicated sentence. Please break into two sentences. We have now split this up into two sentences.

Page 7, line 31. What is A&E? Thank you – we have changed this to Emergency Departments

Page 7, line 56. The section heading is Study Design and Setting, but there is only a single sentence describing the study design. Please provide additional information about the setting. Information about participating GP's is particularly needed. Thank you, we have now added additional information in this section.

Page 8, section 2.4. Please describe the incentive, if any for patients' participation. (There is mention of an incentive in Figure 1, but it should also be included in the text of the protocol.) Thank you, we have now added this information in the text.

Page 8, line 39. There is a typo after "liver" Thank you, this has been updated now.

Page 8, line 57. Additional details need to be included about how many missed weekly and monthly reports results in action being taken. Thank you, additional details have been added.

Page 9, line 3, edit to "They will also be asked..." Thank you, this has been updated now.

Sections 2.12 and 2.13: I do not feel qualified to review these sections.

Page 10, line 34, should it be "type of regimen"? Thank you we have corrected this

From a brief literature search, it appears there are quite a few recent relevant articles the authors may want to consider incorporating.

Thank you, from an updated literature search we found no studies which compare D-mannose alone to either usual care or placebo in a relevant population for our question. We have included an additional reference in the introduction to a study comparing D-mannose in addition to antibiotics, to antibiotics alone. (Kuzmenko et al) The most relevant paper is: Kuzmenko AV, Kuzmenko VV, Gyaurgiev TA. Urologiia. 2019;(6):38-43. Efficacy of combined antibacterial-prebiotic therapy in combination with D-mannose in women with uncomplicated lower urinary tract infection

Reviewer: 3

Wagenlehner Florian

Justus Liebig University Giessen, Germany

Please state any competing interests or state 'None declared':

none

The study protocol is well designed and able to address the research question.

In the sample size calculation paragraph the authors rely to a publication of another group investigating recurrent UTI, where in the control arm only 26% of included patients had at least one UTI in the 6 months follow up.

If the definition of recurrent UTI would be applied, one would expect that 100% of the patients in the control arm would have at least 2 UTIs in the 6 months follow up.

How would this discrepancy impact the study? Thank you, the definition is 2 UTIs in the previous 6 months to qualify for entry into the study. This doesn't mean they will necessarily be expected to have a further 2 UTIs in the following 6 months. We have based the sample size calculation on 26% of the control arm having at least one UTI in the 6 months follow up.

Reviewer: 4

John Stephenson

University of Huddersfield, UK

Please state any competing interests or state 'None declared':

None declared

My review focuses on the statistical aspects of the protocol only. I would like to request the following points of clarification.

1. I was able to reproduce the sample size calculation (section 2.8) when using the Fisher exact test. I assume that is what the authors used to generate their figures. It would have been helpful if they had stated that this was the method they were using, rather than any of the other possible options including options such as uncorrected chi square which are more commonly used than the Fisher test. The reviewer is correct in assuming that Fisher's exact test was used to determine the sample size. We have added the following to the sample size calculation 'To detect this reduction with a 2-sided Fisher's exact test with 90% power and an alpha of 0.05 we would require 203 participants in each arm.'

2. Please can the authors explain why they are comparing treatment groups on the basis of an adjusted risk ratio (section 2.9) rather than an adjusted odds ratio. Please see response to point 3 below

3. Please can they also explain why they are using a log link function (section 2.9) for the primary outcome and not a logit link or similar as I would have expected to see for a proportion. The log binomial model is a binomial generalised linear model with a log link function. It has been chosen to analyse this data as it provides a relative risk rather than an odds ratio, which is often preferred in clinical trials from an interpretation point of view.

4. Section 2.5: "Randomisation will be stratified by GP practice ensuring a balance of the two arms within each practice." Presumably some form of block randomisation will be utilised to "ensure balance" - but no details are given. How large are the blocks? Are block sizes randomised? block randomisation was used, with two block sizes, A&B, randomly selected during sequence construction. A distinct allocation sequence is built for each GP Practice. This has been clarified in the protocol paper. In order to maintain allocation concealment we do not publish the block sizes and they were only known by the IT programmer who sets up the randomisation.

5. It would be interesting to know why it is necessary to stratify by practice. The implication is that there is some key practice-based factor which it is essential to distribute equally across groups. Achieving this comes at a cost (more elaborate analytical methods needed; less uncertainty in pre-determination of allocation. Please could the authors explain why they felt that this procedure was necessary. What is the factor that has to be balanced across groups? With a sample size of over 500, what is the risk of this not happening using blocked, but unstratified randomisation? We considered the inclusion of GP site as a stratification factor and due to the potential differences between practices in socio economic status and deprivation it was decided to include it. Whilst these factors may not directly affect the outcome, this is unknown and therefore it was felt important to ensure balanced allocation within each GP practice, as is usual with primary care trials. This should not affect allocation concealment as the trial is blinded and uses variable block sizes for allocation. We pre specified an analysis which includes site as a random effect.

6. What is the plan for missing data, in particular missing outcome data? I couldn't find that. The researchers state that they will be conducting an ITT analysis but this can be highly problematic with missing data. The primary outcome is obtained from the notes review so we are not expecting much missing data for the primary outcome. The generalised linear mixed model chosen to analyse the primary outcome implicitly accounts for data missing at random, however the data missing mechanism will be explored. Summary statistics will be presented for baseline covariates of those participants who completed and those who were lost to follow-up for the primary outcome. The missing at random assumption will be tested by analysing each baseline covariate in a logistic regression model to determine which (if any) are associated with missingness of the primary outcome, the associated P-value will be reported alongside the summary statistics. Any baseline factors found to be associated with missingness of the primary outcome will be assessed in a sensitivity analysis. A brief explanation has been added to the stats analysis section.

VERSION 2 – REVIEW

REVIEWER	Wagenlehner Florian Justus Liebig University Giessen, Germany
REVIEW RETURNED	22-Sep-2020

GENERAL COMMENTS	The authors have included all raised comments.
--

REVIEWER	John Stephenson University of Huddersfield, United Kingdom
REVIEW RETURNED	07-Oct-2020

GENERAL COMMENTS	I am satisfied that the authors have addressed the majority of my concerns. I have 2 outstanding comments to responses to my comments numbered 3 and 6. 3. I would like to see a stronger justification made for using a log binomial model for the primary outcome, rather than a logit link, which would be a much more common choice for data of this kind. The limit of the justification given is the statement that the log binomial model "provides a relative risk rather than an odds ratio, which is often preferred in clinical trials from an interpretation point of view". However, it could equally well be claimed that OR is also very often preferred to RR, for various very good reasons! There are well-known limitations of RR which the researchers will be aware of. Also, modelling binary data using log binomial regression can lead to estimates lying outside permissible limits, and it is known that confidence intervals for RR from log-binomial models may be under-conservative. Maybe the use of the log link fits in with some theoretical understanding - could this or some other justification that is somewhat stronger than the existing one be added. Otherwise I would restrict it to the analysis of secondary outcomes which are expressed as counts. 6. In their response document the researchers provide some detail about how the missing at random assumption will be tested. But there is much less detail in the manuscript itself. Also, please check whether you are in fact referring to data missing completely at random (MCAR), rather than MAR, as there are no tests available to directly test MAR.
---

VERSION 2 – AUTHOR RESPONSE

Reviewer(s)' Comments to Author:

Reviewer: 3

Reviewer Name: Wagenlehner Florian

Institution and Country: Justus Liebig University Giessen, Germany

Please state any competing interests or state 'None declared': Klosterfrau, Germany

Please leave your comments for the authors below

The authors have included all raised comments.

Thank you.

Reviewer: 4

Reviewer Name: John Stephenson

Institution and Country: University of Huddersfield, United Kingdom

Please state any competing interests or state 'None declared': None declared

Please leave your comments for the authors below

I am satisfied that the authors have addressed the majority of my concerns. I have 2 outstanding comments to responses to my comments numbered 3 and 6.

3. I would like to see a stronger justification made for using a log binomial model for the primary outcome, rather than a logit link, which would be a much more common choice for data of this kind. The limit of the justification given is the statement that the log binomial model "provides a relative risk rather than an odds ratio, which is often preferred in clinical trials from an interpretation point of view". However, it could equally well be claimed that OR is also very often preferred to RR, for various very good reasons! There are well-known limitations of RR which the researchers will be aware of. Also, modelling binary data using log binomial regression can lead to estimates lying outside permissible limits, and it is known that confidence intervals for RR from log-binomial models may be under-conservative.

Maybe the use of the log link fits in with some theoretical understanding - could this or some other justification that is somewhat stronger than the existing one be added. Otherwise I would restrict it to the analysis of secondary outcomes which are expressed as counts.

Thank you for your comments. We agree that logistic regression does have many appealing properties but it is also not without its problems, including interpretation and the fact that it can give an estimate which is driven away from null when adjusted due to issues of non-collapsibility. We have specified in our statistical analysis plan that if the log binomial model fails to converge we will use a generalised linear model with a Poisson distribution, a log link function, and robust standard errors to analyse the data, and that if this also fails to converge we will use a logistic regression model. The decision to use the proposed methods for the primary analysis was based on our previous experience of reviewers and journals requesting relative risks rather than odds ratios and as such it is very hard to pre define an analysis to be used.

6. In their response document the researchers provide some detail about how the missing at random assumption will be tested. But there is much less detail in the manuscript itself. Also, please check whether you are in fact referring to data missing completely at random (MCAR), rather than MAR, as there are no tests available to directly test MAR.

Sorry this is our mistake, we will not be 'testing' the MAR assumption but will be testing which baseline covariates are associated with missingness, so that these can be included in a sensitivity analysis. We have now included more detail in the analysis section.